# Analysis of LTE-M Adjacent Channel Interference in Rail Transit

**DOI:** 10.3390/s22103876

**Published:** 2022-05-20

**Authors:** Hao Fu, Xiaoyong Wang, Xuefan Zhang, Asad Saleem, Guoxin Zheng

**Affiliations:** 1Key Laboratory of Specialty Fiber Optics and Optical Access Networks, Shanghai University, Shanghai 200444, China; 18800201525@163.com (H.F.); 10002461@t.shu.edu.cn (X.Z.); 2CASCO Signal Ltd., Shanghai 200071, China; wangxiaoyong@casco.com.cn; 3Key Laboratory of Antennas and Propagation, Shenzhen University, Shenzhen 518060, China; asadalvi64@szu.edu.cn

**Keywords:** rail transit, LTE-M, adjacent channel interference, isolation degree

## Abstract

Long Term Evolution-Metro (LTE-M), as a special communication system for train control, has strict requirements on adjacent channel interference (ACI). According to the 3rd Generation Partnership Project (3GPP) protocol of the European Telecommunications Standards Institute (ETSI) standards, this paper presents the required isolation degree for LTE-M systems to resist ACI. Aiming at the scenario of leaky cable transmission and antenna transmission adopted by the underground LTE-M system of the subway, the isolation degree required for LTE-M system deployment is deduced by combining the channel description with the principle of ACI. For the coexistence of a LTE-M system and an adjacent cellular system in a subway ground scenario, the Monte-Carlo (MC) method is used to simulate several conceivable scenarios of the LTE-M system and the adjacent frequency cellular system. In addition, the throughput loss of the LTE-M system is estimated by considering signal to interference plus noise ratio (SINR). Simulation results demonstrate that adjacent frequency user equipment (UE) has negligible small interference with the LTE-M underground system when using the leaky cable radiation pattern, whereas for the LTE-M ground system, the main interference comes from the adjacent frequency UE to the LTE-M base station (BS). Finally, interference avoidance solutions are presented, which can be utilized as a reference in the design and deployment of LTE-M systems in the rail transit environment.

## 1. Introduction

Long-Term Evolution (LTE) is a fourth-generation communication technology with exceptional mobile access competence and high system dependability in high-speed urban rail transportation [1]. Therefore, the Long-Term Evolution-Metro system, which is based on LTE technology, is proposed as the primary communication method for urban rail transit systems [2] and carries the wireless network of the communication-based train control (CBTC) system, ensuring that CBTC information has the highest priority when transmitted in the LTE-M system. LTE-M has become extremely popular since the advent of the vehicle-ground information transmission service and the train entertainment information service [3,4]. In 2015, the Chinese Ministry of Industry and Information Technology announced that the frequency range 1785–1805 MHz is intended for dependable communications in the urban rail transit industry in order to improve rail transit security [5]. However, in the adjacent frequency bands of LTE-M system, there are frequently other communication frequency bands that can cause interference to LTE-M systems [6]. When the interference is severe, the train is forced to brake suddenly. Several emergency brake accidents have occurred in Shenzhen and Beijing, China, due to the interference of user equipment [7]. Therefore, it is critical to look into the causes of interference and design strategies for removing it. The following two methods are commonly used in interference analysis. The deterministic method, which is based on an estimate of the minimum coupling loss (MCL), is one method for estimating the need for isolation for worst-case interference. The other one is based on the Monte-Carlo statistical method, which is used to model random processes and obtain reliable data through a sufficient number of simulations. This method is appropriate for a more precise assessment of interference coexistence between systems in the case of multiple interferences [8].

### 1.1. Related Literature

Many studies on interference have been conducted in recent years, mainly including interference between 4G and 5G systems [9,10,11,12], 4G and 4G communication systems [13,14,15], satellite service ground station systems and 5G systems [16,17,18,19,20], and LTE and broadcasting systems [21,22,23]. In 5G new radio (NR) frequency-division duplexing (FDD) systems that coexist with LTE systems, simulation study is executed to explore the required isolation degree under common and non-common station of the base station [9,10,11]. The results show that non-common station scenarios require a higher isolation degree. On the basis of [9,10,11], the authors of [12] investigated the interference coexistence of LTE and NR systems using dynamic spectrum sharing (DSS) technology and discovered that LTE neighbor cells have a stronger impact on NR cells. The deterministic analysis method is utilized to investigate coexistence interference between LTE FDD and Time Division Long Term Evolution (TD-LTE) systems, as well as between TD-LTE and Time Division-Synchronous Code Division Multiple Access (TD-SCDMA) systems in [13,14,15]. The main emphasis is devoted towards the isolation of the coexistence of base stations under common address, common coverage, and the safe distance between base stations. In [16,17,18,19,20], the minimum coupling loss method is used to evaluate the interference of adjacent channels between different service earth stations, 5G base stations, and user equipment. The results show that, the source’s interference power should be lowered, and more frequency separation needs to be used to assure system coexistence. In [21,22,23], the authors focus on the analysis of ACI between the LTE mobile systems and Digital Video Broadcasting-Terrestrial (DVB-T) systems, with the main conclusion that increasing the separation distance between LTE BS and DVB-T reduces interference between adjacent channels, and the minimum separation distance decreases with the presence of a guard band.

The purpose of a coexistence study is to enable the compatibility between adjacent channel systems through the interference analysis. If the coexistence of the systems cannot be ensured using the methods described in the literature, a range of approaches are utilized to reduce system interference and ensure coexistence. In [24,25], a novel approach for suppressing ACI induced by RF front-end non-linearity is described, which involves digital frequency domain filtering of adjacent signals to remove out-of-band emissions from the received signal. In [26], the power control scheme is proposed to mitigate the impact of ACI on vehicle-to-vehicle broadcast communication. In [27], a measurement data-driven machine learning paradigm is proposed to set power control parameters for optimal uplink interference management of a LTE system. In [28], the impact of co-channel interference on a Global System for Mobile Communications-Railway (GSM-R) system is evaluated for the signal received power threshold that affects the performance of a GSM-R network. In [29], an optimizing model is proposed to maximize the receiving SINR to suppress the jamming in railway wireless communication systems. As for the abnormal train-ground communication caused by complex environmental factors, electric arc dynamics, and electromagnetic noise along the subway, Hammi et al.’s study [30] shows that the main source is the transient electromagnetic interference generated by the sliding contact between catenary and pantograph, which has little impact on the communication system. The authors of [31] propose a method to improve the anti-electromagnetic interference ability of the entire railway vehicle from the grounding and wiring aspects. In [32,33], the authors proposed that signal interference can be solved by establishing a shielding network or adding a filter to the base station.

### 1.2. Motivation and Contributions

Existing research focuses on interference between LTE and other communication systems in various frequency bands, with the majority of them examining interference between LTE-FDD, TD-LTE, NR-FDD, TD-SCDMA, and DVB-T system cells caused by base stations. These systems are not the main communication means of urban rail transit system, and their frequency band is not within the range of 1.8 GHz. In addition, compared with the base station, the user terminals are widely distributed, and the location is uncertain, which poses a greater threat to train control. Currently, the analysis approach for a 1.8 GHz LTE-M system subjected to ACI still lacks related deterministic calculation and system-level simulation. Therefore, in this paper, the deterministic model under different interference scenarios for a LTE-M underground system is derived. Moreover, the wireless channel propagation model between different communications objects is presented for LTE-M system; the minimum isolation degree is obtained through interference link calculations. The LTE FDD cellular topology and LTE-M linear topology models, as well as the power control model and throughput calculation process, are all established for the LTE-M ground system. The minimum isolation degree is obtained by changing the simulation parameters. Finally, the anti-interference recommendations for urban rail transit systems are provided as part of the interference reduction strategies.

### 1.3. Article Structure

The rest of this paper is organized as follows. In Section 2, the interference principle of a LTE-M at adjacent frequency bands is introduced in detail. Section 3 analyzes the interference of a LTE-M underground system and provides critical analysis. In Section 4, the interference of a LTE-M ground system is analyzed and simulation results are given. Finally, Section 5 provides the conclusion of this paper.

## 2. LTE-M Adjacent Channel Interference

### 2.1. Frequency Band Division

Recently, rail transits have used a wide range of civil and private wireless communication systems. As their frequency and space localization are similar to the LTE-M, the adjacent channel interference exists between them. In addition, private wireless communication is principally based on the LTE-M system, which operates in the 1785–1805 MHz frequency range, whereas the civil communication systems of other operators are distributed in their adjacent frequency bands [34]. The specific distribution of LTE-M adjacent frequency bands is shown in Table 1.

Table 1 reveals that the two upper adjacent channel interference sources are both immovable base stations. As opposed to the base station with a relatively fixed location, the lower adjacent channel interference source is user equipment, the distribution of which is fairly random, and the position can be changed at any time. Once the interfering user is very close to the LTE-M system, the interference will become severe. Therefore, this paper focuses on the interference between lower adjacent frequency bands based on the LTE FDD UE and LTE-M systems.

### 2.2. Principle of Adjacent Channel Interference

ACI is mainly determined by the characteristics of the transmitter and receiver filters. In Figure 1, A is the fraction of the interference power of the transmitter entering the targeted frequency band due to out-of-band radiation characteristics, and B is the fraction of the interference signal power that the receiver can receive in the adjacent channel due to imperfect filtering characteristics.

Adjacent channel leakage ratio (ACLR) is the ratio of the filtered mean power centered on the assigned channel frequency to the filtered mean power centered on adjacent channel frequency, which is used to measure the out of band propagation characteristics of the transmitter and can be shown as
(1)$$\mathrm{ACLR}=\frac{P}{A}=\frac{1}{\int_{f_{imax}+\Delta f}^{+\infty }E_{f_{i}}\left(f\right)df}$$
where $E_{f_{i}}\left(f\right)$ is the power spectral density of the transmission filter of LTE FDD system, $f_{imax}$ is the nominal upper cut-off frequency of the filter, and Δf is the guard interval between LTE FDD and LTE-M.

Adjacent channel selectivity (ACS) represents the ratio of the receive filter attenuation on the targeted channel to that on the adjacent interference channel, which is used to measure the receiver’s performance in the targeted band, can be shown as
(2)$$\mathrm{ACS}=\frac{P}{B}=\frac{1}{\int_{-\infty }^{f_{omin}-\Delta f}E_{f_{o}}\left(f\right)df}$$
where $E_{f_{o}}\left(f\right)$ is the power spectral density of the receiving filter of LTE-M system, $f_{omin}$ is the nominal lower cut-off frequency of the filter.

Adjacent channel interference ratio (ACIR) is the ratio of the total interference power received by the targeted signal receiver to the total interference power sent by the adjacent signal transmitter [35]. The linear value of ACIR can be expressed as
(3)$$\mathrm{ACIR}=\frac{P}{A+B}=\frac{P}{\frac{P}{\mathrm{ACLR}}+\frac{P}{\mathrm{ACS}}}=\frac{1}{\frac{1}{\mathrm{ACLR}}+\frac{1}{\mathrm{ACS}}}$$

The value of ACIR (dB) derived from Equation (3) can be re-written as
(4)$$\mathrm{ACIR}\;\left(\mathrm{dB}\right)=-10\mathrm{lg}(10^{-\frac{\mathrm{ACLR}}{10}}+10^{-\frac{\mathrm{ACS}}{10}})$$

LTE-M base station and LTE-M vehicle terminal (TE) has tolerable interference signal noise power threshold $I_{threshold}$ in dBm; once the noise power exceeds this threshold, it will block the reception of the targeted signal [36]. The threshold is related to the performance of the receiver, which is specifically manifested in the sensitivity loss of the receiver S. $I_{threshold}$ can be written as
(5)$$I_{threshold}=P_{noise}+10\mathrm{lg}(10^{\frac{S}{10}}-1)$$

According to [37], the sensitivity loss $S_{BS}$ of base station receiver is generally 0.8 dB, and the sensitivity loss $S_{TE}$ of the vehicle terminal receiver is generally 3 dB. The receiver noise floor power $P_{noise}$ is related to the system receiving bandwidth (BW) and the receiver noise value $N_{F}$ [38], as shown below
(6)$$P_{noise}=-174\;\mathrm{dBm}+10\mathrm{lg}\mathrm{BW}+N_{F}$$
where the thermal noise floor level throughout a 1 Hz bandwidth at 27 °C is −174 dBm and the noise coefficient $N_{F\_BS}$ of the base station receiver is 5 dB and that of the vehicle terminal receiver $N_{F\_TE}$ is 9 dB [39].

This paper selects the transmission link with the most serious interference; that is, one with the highest transmission power $P_{Imax}$ and the highest transceiver antenna gain $G_{TR}$. In the transmission process, the transmitter transmits the interference signal, and the signal arrives at the receiver after passing through the transmitting antenna, spatial wireless transmission, channel attenuation, and receiving antenna. The transmission loss occurs when the actual received interference signal power meets $I_{threshold}$, which represents the MCL [40]; that is, the minimum spatial isolation, as given follows
(7)$$\mathrm{MCL}=P_{Imax}+G_{\mathrm{TR}}-\mathrm{ACIR}-I_{threshold}$$

Suppose the path loss of signal transmission in the interfering LTE FDD UE and the LTE-M system is L, when L>MCL, the path loss exceeds the minimum spatial isolation, and when L<MCL, the path loss meets the minimum spatial isolation requirement. In the case of L=MCL, this paper uses different wireless channel propagation models to estimate the minimum safe distance between different systems.

Table 2 shows the steps involved in calculating the LTE-M system’s interference parameters, where $P_{noise}$ can be obtained by substituting different system receiving bandwidths and the receiver noise values into Equation (6), then substituting the $P_{noise}$ and different sensitivity loss into Equation (5) to obtain $I_{threshold}$. Since both the transmitter and the receiver filters are non-ideal, the frequency band interval and the received signal bandwidth is related to the component of transmitted power falling into the adjacent receiver bandwidth and the attenuation of transmitted power by the receiver filter. According to the spectrum emission mask in [41], as shown in Table 3, the UE’s spectrum emission mask is applied to the frequency of the out of band emission (Δf_OOB_), starting from the edge of the assigned channel bandwidth and the value of ACLR and ACS under different guard band intervals and different receiving channel bandwidths can be obtained by the method of integration. Then, the value of ACLR and ACS can substituted into Equation (4) to obtain the value of ACIR. According to [41], the maximum transmitted power of UE is 23 dBm, the antenna gain of LTE-M BS is 15 dBi, and the antenna gain of LTE-M TE is 0. Moreover, the MCL under different interference links can be obtained by substituting $P_{Imax}$, $G_{TR}$, ACIR, and $I_{threshold}$ into Equation (7).

## 3. Interference Analysis of LTE-M Underground System

### 3.1. Interference Analysis When LTE-M Uses Directional Antenna Radiations Pattern

When the LTE-M system uses the directional antenna radiations pattern, the scene of interference is shown in Figure 2.

The source of interference is LTE FDD UE, which may be located in the platform area or inside the carriage, and the sources of victimization are LTE-M BS and LTE-M TE. This paper focuses on two types of interference, one is LTE FDD UE interfering with the LTE-M BS (UE-BS), and the other is LTE FDD UE interfering with LTE-M TE (UE-TE).

The path loss of signal in the transmission process is greatly affected by the frequency, environment, distance, antenna height, and some other factors. Selection of an appropriate propagation model is a necessary condition to analyze the interference. Assuming that the distance between UE and the BS is less than 100 m and the UE is located in the platform area, the propagation model can be approximately conceived as the free-space loss according to Equation (8), f represents the frequency of electromagnetic wave in MHz, and d represents the spatial distance between transceivers.
(8)$$L_{free}=32.44+20\mathrm{lg}\left(d\right)+20\mathrm{lg}\left(f\right)$$

When the UE is located inside the carriage, the UE needs to penetrate the carriage to reach the receiving end. The Keenan-Motley model [42], which was derived from the free space propagation model, accounts for penetration loss in the propagation environment, given as follows
(9)$$L_{k}=L\left(d_{0}\right)+20\mathrm{lg}\left(\frac{d}{d_{0}}\right)+kF\left(k\right)$$
where $L(d_{0})$ is the path loss of the free space with a reference distance of $d_{0}$, F(k) is the reference value of the penetration loss, *k* is the number of layers of penetration. In this paper, the penetration is 1 layer, and the penetration loss is 8 dB.

Assuming that the distance between UE and the BS is greater than 100 m, the vehicle-mounted propagation model can be estimated as
(10)$$L_{\mathrm{UE}-\mathrm{BS}}=40\left(1-0.004\Delta h_{\mathrm{BS}}\right)lg\left(d\right)-18-18lg(\Delta h_{\mathrm{BS}})+21lg(f)+80$$
we assume this value as 5 m, f represents the frequency of the electromagnetic wave in MHz, and d represents the spatial distance between transceivers.

When the UE is located in the platform area and the distance between the UE and the TE is less than 50 m, the free-space loss model (8) can be used to examine the entire transmission process. When the UE is located inside the carriage, the transmission model uses the Keenan-Motley model as in Equation (9). Assuming that the separation distance between UE and the TE is greater than 50 m, the transmission model uses the Xia.h model [43], such as,
(11)$$L_{\mathrm{UE}-\mathrm{TE}}=-10lg{\left(\frac{\lambda }{4\pi d}\right)}^{2}-10lg\left[\frac{\lambda }{2\pi^{2}r}{\left(\frac{1}{\theta }-\frac{1}{2\pi +\theta }\right)}^{2}\right]-10lg\left[{\left(\frac{b}{2\pi d}\right)}^{2}\frac{\lambda }{\sqrt{\Delta h_{m}{}^{2}+b^{2}}}{\left(\frac{1}{\phi }-\frac{1}{2\pi +\phi }\right)}^{2}\right]$$
where $r=\sqrt{{\left(\Delta h_{m}\right)}^{2}+x^{2}}$, $\theta =\tan^{-1}(\frac{\Delta h_{m}}{x})$, $\phi =\tan^{-1}(\frac{\Delta h_{b}}{b})$. λ is the carrier wavelength, x represents the horizontal spacing between the terminal and the scattered edge, typically 15 m, b represents the average distance between constructions, with a typical value of 80 m, $\Delta h_{m}$ represents the average height difference between the base station antenna and the terminal, which is 3.5 m, and $\Delta h_{b}$ represents the height difference between the base station antenna and the track, and considered as 5 m.

Under the condition of L=MCL, the minimum safe distance under different conditions can be achieved after substituting MCL into Equations (8)–(11), as shown in Table 4 and Table 5. By taking the transmission power and the antenna gain as dependent variables, the relationship between the transmitting power of interference signal, the antenna gain of BS, the received signal BW, the guard band interval, and the minimum safe distance in different links can be obtained, as shown in Figure 3 and Figure 4.

When the transmitting power of interfering signal or antenna gain of BS increases, the minimum safe distance also increases, indicating that lowering them can decrease interference of the LTE-M system. When the received signal BW increases, the minimum safe distance decreases gradually, because $I_{threshold}$ increases with the increase in the received signal BW. The anti-interference ability of LTE-M BS is improved by 2.88 dB when the received signal BW is 10 MHz compared with 5 MHz, as computed from the MCL in Table 2. When other factors are the same, the minimum safe distance with 5 MHZ guard interval is smaller than without guard interval. After the 5 MHz guard interval is implemented, the anti-interference capability of the LTE-M BS improves by 13.65 dB when compared to no guard interval, as computed from the MCL in Table 2.

Similarly, the LTE FDD UE interfering with LTE-M TE is obtained as shown in Figure 5, and it can be noticed that the main trend is similar to that of LTE FDD UE interfering with LTE-M BS. Regardless of whether the UE is in the platform or inside the carriage, the minimum safe distance is very small, which means that the interference of the UE to the LTE-M system is negligibly small in this case. When all other conditions are equal, LTE-M TE’s anti-interference capability improves by 2.84 dB when the received signal BW is 10 MHz, compared with 5 MHz. It can also be noticed that the anti-interference capability of the LTE-M TE is improved by 13.5 dB after the 5 MHz band guard interval is added compared to when no guard interval is used.

### 3.2. Interference Analysis When LTE-M Uses Leaky Cable Radiations Pattern

When the LTE-M system uses the leaky cable radiations pattern, the complete scene of interference is shown in Figure 6.

The transmission channel based on leaky cable can be divided into the inner channel and the spatial channel. The spatial channel can be conceived as the wireless fading channel, where the signal travels from the slot to the UE. The longitudinal loss inside the leaky cable can be abstracted as linear power attenuation related to the slot period of the leaky cable [44], so the attenuation model can be obtained:(12)$$a_{i}=\alpha \cdot p\cdot i\;$$
where $a_{i\;}$ represents the amplitude of radiation signals from i slots, a represents the attenuation factor during longitudinal transmission of leaky cable, and p is the slot period, as shown in Figure 7.

When the interfering signal interferes with the Remote Radio Unit (RRU) of the LTE-M BS through the leaky cable, MCL can be estimated as
(13)$$\mathrm{MCL}=P_{Imax}-\mathrm{ACIR}-I_{threshold}$$
(14)L=L1+L2+L3
(15)$$L2=L_{2m}+32.44+20\mathrm{lg}\left(d1\right)+20\mathrm{lg}\left(f\right)$$
where L1 is the longitudinal transmission loss of leaky cable, L2 is the coupling loss of leaky cable, d1 is the vertical distance between UE and leaky cable, and L3 is the sum of losses due to feeder, splitter, train penetration, and body penetration. The train penetration loss is considered when the source of interference is located inside the train carriage. Considering the most serious interference; that is, the power of the interference signal fed through the last slot of the leaky cable, the longitudinal transmission loss can be obtained by substituting it into Equation (12): L1=4 dB/100 m×0.25 m×1=0.01 dB. According to the Table 6, L3=0.9 dB+1.4 dB+4 dB+2 dB=8.5 dB. In the case of L=MCL, the minimum value of L2 can be calculated by Equations (13)–(15) to obtain the minimum safe distance between UE and the leaky cable, as shown in Table 6.

Taking the power of the interference signal and L3 as a dependent variable, the relationship between interference signal transmission power, L3, the received signal bandwidth, and the minimum safety distance (d1) in different environments can be obtained according to Equations (13)–(15), as shown in Figure 8.

Similarly, the interference to the LTE-M system can be reduced by reducing the transmitting power. When L3 increases, the minimum safe distance gradually decreases. Because the larger L3 is, the greater the attenuation of interference signal will be, and the interference to the LTE-M system will gradually decrease. It can be seen from Table 6 that the maximum of the minimum safe distance in four given cases is 0.85 m, and the distance between the train and the leaky cable beside the tunnel wall is 2 m, so there is no adjacent channel interference, even in the most severe cases when the interference signal was fed through the last slot of the leaky cable.

## 4. Interference Analysis of LTE-M Ground System

Aiming at the ground interference of the LTE-M system, this paper uses the Monte Carlo method to simulate and analyze throughput, which is the main evaluation index of system performance.

### 4.1. Network Topology

In this paper, the regular hexagonal macro cellular network is considered for the LTE FDD system topology structure. The macro cellular network topology structure is shown in Figure 9, where 1 represents the central cell, each cell includes three identical sectors and the cell base station adopts a 65° sector directional antenna. The simulation area simulates the layout of 19 regular hexagonal cells at two floors. The radius *r* of the cell is 250 m, the radius *R* of the macro cell is 433 m, and the inter-site distance (*ISD*) between the center base stations of the adjacent cell is 750 m. When simulating interference, wrap around the edge causes each cell to have the influence of at least two layers of peripheral cells.

The LTE-M system model is not the same as the LTE FDD cellular network, and there is currently no standard protocol for it. Therefore, this paper establishes a linear topology model for LTE-M train operation system based on actual track size and train parameters. The given model is based on a linear train track. The train is located randomly on the linear track, and the vertical distance from LTE-M BS is no more than 500 m, and the LTE-M BS is set at a horizontal distance of 30 m from the track. The coverage radius of a single base station is 500 m. The system topology structure is shown in Figure 9.

Offset *D* indicates the distance between the interferer system and the victim system. Figure 10 represents the offset between the two systems. When *D* = 0, it indicates the common location of the two systems, and when *D = R = ISD/*√*3,* it indicates that the interfere system is at the cell edge of the victim system.

### 4.2. ACLR Model

In the uplink, ACIR depends on the ACLR of the UE. The value of the ACLR model varies according to the interval of the adjacent channel band [39]. For a system with any bandwidth, a user in the downlink occupies only one resource block (RB), while all RBs in the uplink are evenly distributed to all users in the scheduling system. There are 50 RBs in 10 MHz bandwidth. If five user terminals are connected in uplink, the number of RBs occupied by a single user is 10 and the bandwidth $B_{A}$ is the number of RBs occupied by a single user multiplied by the LTE RB width of 180 KHz, where X serves as the step size for simulations, X = … −10, −5, 0, 5, 10… dB, as shown in Table 7.

### 4.3. Power Control

The uplink users change the transmitted power through the power control model [39], given as follows
(16)$$P_{t}=P_{max}*min\left\{1,max\left[R_{min},{\left(\frac{PL}{PL_{x-ile}}\right)}^{\gamma }\right]\right\}$$
where $R_{min}=P_{min}/P_{max}$ is the minimum power reduction ratio to prevent UEs from good channel conditions from transmitting with very low power, PL represents the path loss from the UE to the BS, and its propagation model is given in Equation (10).$\;PL_{x-ile}$ is the *x*-percentile path loss value, which is used to ensure the *x* percent of UEs with highest propagation loss transmission will transmit at $P_{Imax}$. Additionally, 0<γ≤1 is the balancing coefficient of UE under different channel conditions. The power control parameter sets are shown in Table 8.

### 4.4. Throughput Calculation

Taking LTE FDD UE interference with LTE-M BS as an example, the SINR can be calculated as follows: (17)$${\mathrm{SINR}}_{i}=\frac{P_{t\cdot i}*G_{i}*PL_{i}}{\sum_{k}^{M}\sum_{j}^{N}P_{t\cdot kj}*G_{kj}*PL_{kj}{*\mathrm{ACIR}}_{kj}+P_{noise}}$$
where ${\mathrm{SINR}}_{i}$ represents the received signal to interference noise ratio of the *i*-th channel of the base station receiver, $P_{t\cdot i}$ is the transmission power of the LTE-M TE of the *i*-th channel, $G_{i}$ is the sum of transmitting and receiving antenna gain between the LTE-M TE and LTE-M BS of the *i*-th channel, $PL_{i}$ is the propagation loss between the LTE-M TE of the *i*-th channel and the receiving LTE-M BS, $P_{t\cdot kj}$ represents the transmission power of the *j*-th FDD UE in the *k*-th cell, $G_{kj}\;$ represents the sum of the antenna gains of the *j*-th FDD UE in the *k*-th cell and the LTE-M BS, $PL_{kj}$ is the propagation loss of the *j*-th FDD UE in the *k*-th cell reaching the LTE-M BS, and its propagation model is Equation (10). ${\mathrm{ACIR}}_{kj}$ is the ACIR value of the *j*-th FDD UE of the *k*-th cell and the LTE-M BS, *N* is the number of cell users, *M* is the number of FDD cells, and $P_{noise}$ is the receiver noise floor power.

The approximate throughput is obtained by Shannon mapping expression [39], as given below
(18)$$\mathrm{TP}=\{\begin{array}{c} 0\;\mathrm{SINR}<{\mathrm{SINR}}_{min}\\ \alpha \times S\left(\mathrm{SINR}\right)\;{\mathrm{SINR}}_{min}<\mathrm{SINR}<{\mathrm{SINR}}_{max}\\ {\mathrm{TP}}_{max}\;\mathrm{SINR}>{\mathrm{SINR}}_{max} \end{array}$$
where *S* is Shannon bound as $S\left(\mathrm{SINR}\right)=\log_{2}\left(1+\mathrm{SINR}\right)$ bps/Hz, α indicates attenuation factor, ${\mathrm{SINR}}_{min}$ is the lower limit of SINR, ${\mathrm{TP}}_{max}$ is the throughput upper limit, ${\mathrm{SINR}}_{max}$ is the SINR when the throughput limit is obtained. For the given system, the LTE interference link parameters are shown in Table 9.

According to [39], simulation results with interference shall be expressed as a percentage of throughput reduction compared to simulation results without external interference, with the victim system’s throughput loss not exceeding 5%. Therefore, in this paper, 5% relative throughput loss is used as the evaluation criterion of the maximum interference of the external system After each simulation, the SINR on each RB is counted. According to the Shannon mapping method, the SINR is mapped with the throughput of each RB, and the system throughput of one simulation is accumulated. The average throughput $TP_{loss}\;$ of the system is obtained by averaging multiple simulations, given as follows
(19)$${\mathrm{TP}}_{loss}=1-\frac{{\mathrm{TP}}_{multi}}{{\mathrm{TP}}_{single}}$$
where ${\mathrm{TP}}_{single}$ is the average throughput of a single LTE-M system and ${\mathrm{TP}}_{multi}$ is the average throughput of the LTE-M system when interfered by LTE FDD system.

### 4.5. Simulation Results

This paper uses the snapshot method to sample the system running process. After each snapshot, the positions of the terminals are changed randomly, and the same simulation algorithm is executed as the last time, and, finally, the sampling results at all snapshot moments are statistically analyzed. MATLAB software is utilized to generate simulation results, and the simulation parameters for the proposed system are as follows (Table 10).

In order to explore the factors affecting system performance, the required ACIR values are obtained by changing different simulation parameters. When the number of interfering users is 50 and the power control parameter Set 1 is selected, the offset between base stations is changed, and the simulation result of throughput loss is shown in Figure 11. When the offset between base stations is D, the throughput loss changes significantly under different interfering users and different power control parameters as shown in Figure 12.

It can be seen from the simulation results that when the offset *D* between base stations is constant, the throughput loss of LTE-M BS and TE decreases with the increase in ACIR. When ACIR is fixed for UE interfering BS, the higher the offset between BS, the greater the throughput loss of BS. Due to the same power control parameters, the distance between the edge user of the LTE FDD system and the LTE-M BS decreases as the offset increases, resulting in increased interference to the LTE-M BS. The outcome is the opposite for UE interfering TE; due to the randomness of the UE distribution, there is no noticeable difference between the curves, and when the ACIR is more than 35 dB, the throughput loss of the LTE-M system is below the 5% threshold. As a result, LTE FDD UE has negligible impact on LTE-M TE. Table 11 shows the required ACIR values to keep the average relative throughput loss below 5% when the distance between the systems is 0, 0.5 *R*, and *R*.

The simulation results show that when the number of users and ACIR are fixed, the throughput loss of Set 1 is higher than that of Set 2. From the power control model, it can be seen that the ${\mathrm{PL}}_{x-\mathrm{ile}}$ of Set 1 is smaller than that of Set 2, so the transmitted power of Set 1 is larger than that of Set 2, and the throughput loss of the LTE-M system is also larger. When the power control parameter sets and ACIR are fixed, the larger the number of users is, the larger the throughput loss will be. Since the number of resource blocks corresponding to each bandwidth length is constant, the greater the number of LTE-FDD users, the higher the overall interference power and the greater the LTE-M system’s throughput loss can be achieved. Table 11 shows the required ACIR values when using different parameter sets and the number of users to keep the average relative throughput loss below 5%.

In general, the offset between two systems should be minimized during their deployment to avoid the occurrence of interfering system construction near the disturbed system’s cell boundary. For the selection of power control parameters, Set 2 can be adopted for power control in places with dense users and Set 1 is suitable for places with sparse users to reduce link interference while fulfilling the full power transmission needs of most users.

## 5. Conclusions

In this paper, different analytic approaches are considered to investigate the interference for the LTE-M underground and ground systems, respectively. The deterministic results obtained by using the MCL method for the LTE-M underground system show that when LTE-M adopts the leaky cable radiation pattern, the LTE FDD UE has negligible small interference with LTE-M system. When LTE-M uses the directional antenna radiations pattern, the LTE FDD UE interfering with the LTE-M system requires a certain safe distance and some measures to avoid interference, such as frequency band isolation, space isolation, changing the transmission power, antenna gain, and received signal bandwidth. In this case, the interference of LTE FDD UE to LTE-M TE is negligibly small. When LTE FDD UE interferes with LTE-M BS, raising the received signal bandwidth by 5 MHz, it improves the anti-interference capability of LTE-M by 2.88 dB, and adding the 5 MHz guard interval improves the anti-interference capability of LTE-M by 13.65 dB. The throughput loss of the LTE-M ground system is simulated using the MC method, and the simulation results show that the ACIR required by LTE FDD UE to interfere with LTE-M TE is typically less than 35 dB, with nearly no interference caused. The main interference comes from the LTE FDD UE interfering with LTE-M BS. The anti-interference capability improves by around 5.75 dB when the number of users is reduced by 50, and anti-interference performance improves by about 3.96 dB when Set 1 is chosen over Set 2. Moreover, it is shown that specific steps, such as lowering the offset *D* and selecting appropriate power control parameters based on the number of UE, must be taken in order to reduce throughput loss.

## Figures and Tables

**Figure 1 sensors-22-03876-f001:**
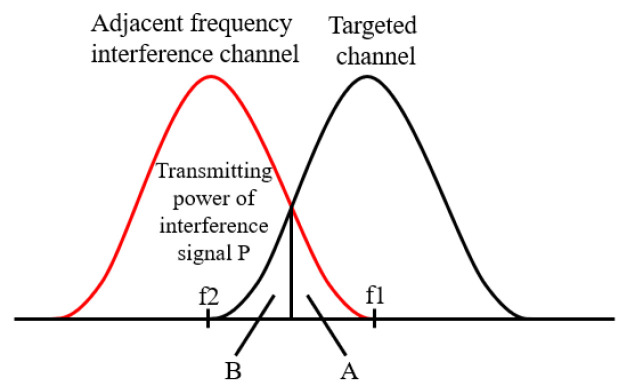
Power distribution of transmitting and receiving signals.

**Figure 2 sensors-22-03876-f002:**
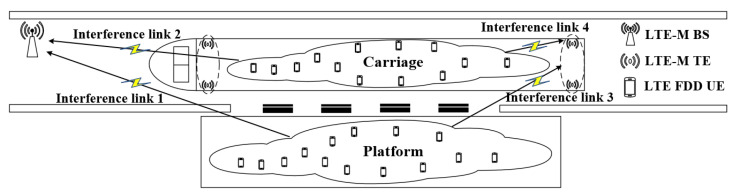
Interference analysis of LTE-M with directional antenna radiations pattern.

**Figure 3 sensors-22-03876-f003:**
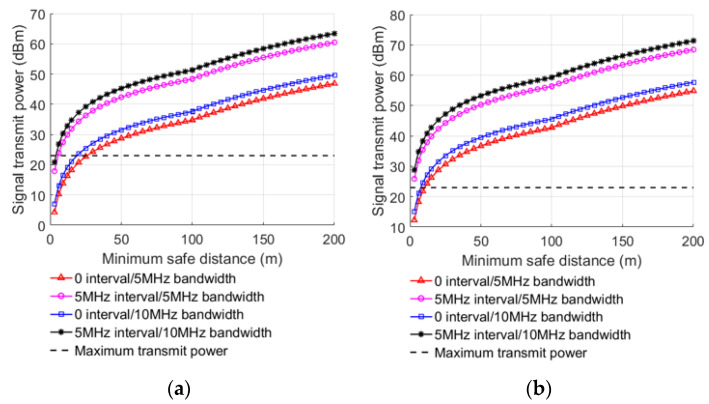
Minimum safe distance with the transmit power of interfering signal at different UE positions when UE interfering with BS: (**a**) platform, (**b**) carriage.

**Figure 4 sensors-22-03876-f004:**
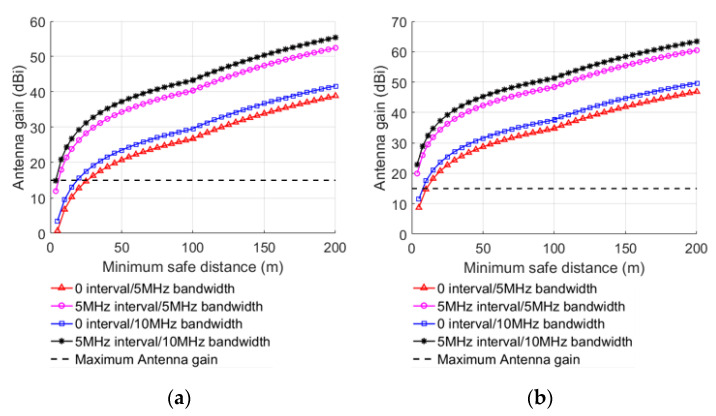
Minimum safe distance with the antenna gain at different UE positions when UE interfering with BS: (**a**) platform, (**b**) carriage.

**Figure 5 sensors-22-03876-f005:**
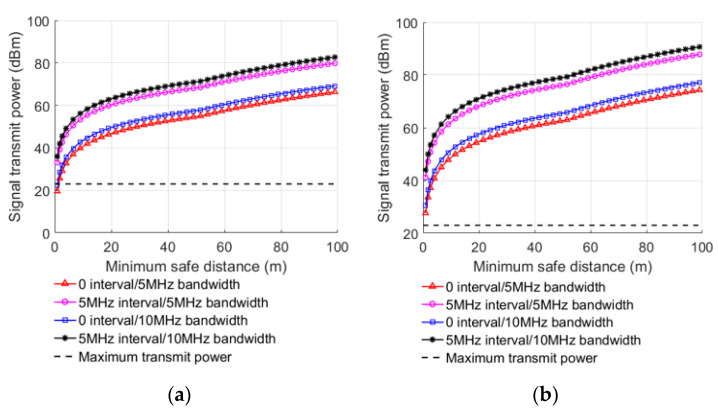
Minimum safe distance with the transmit power of interfering signal at different UE positions when UE interfering with TE: (**a**) platform, (**b**) carriage.

**Figure 6 sensors-22-03876-f006:**
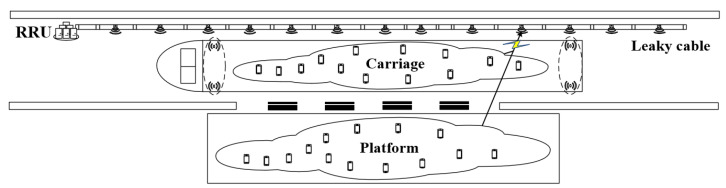
Interference analysis of LTE-M with leaky cable radiations pattern.

**Figure 7 sensors-22-03876-f007:**
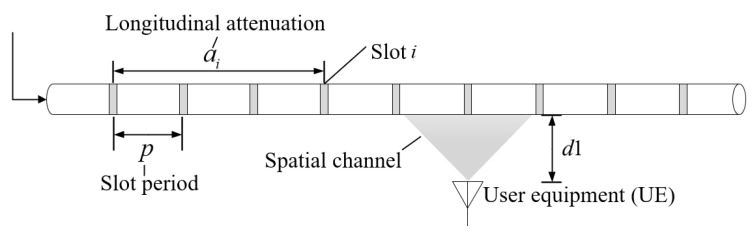
Channel transmission in leaky cable.

**Figure 8 sensors-22-03876-f008:**
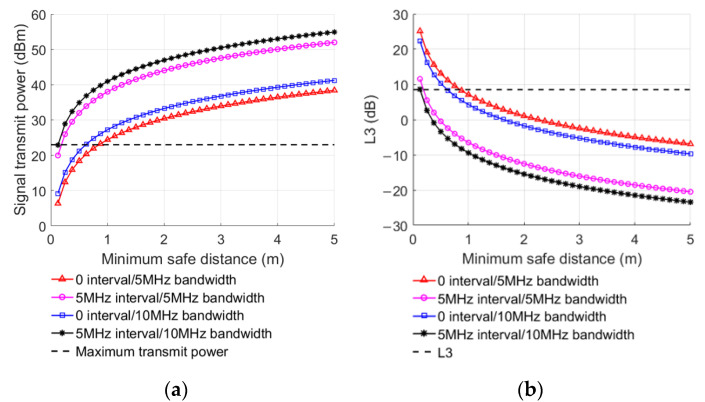
Influencing factors when UE interfering with BS: (**a**) transmitting power of interference signal, (**b**) L3.

**Figure 9 sensors-22-03876-f009:**
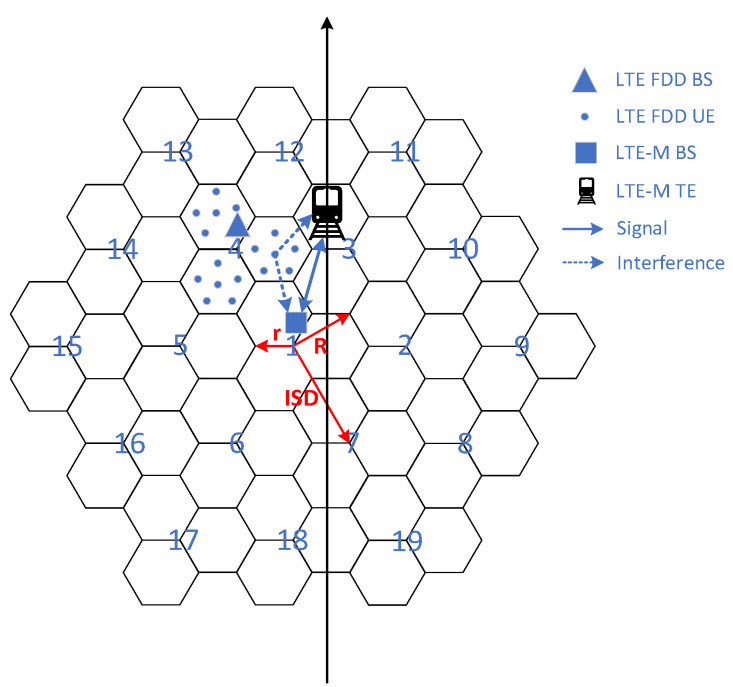
Interference scenario network topology.

**Figure 10 sensors-22-03876-f010:**
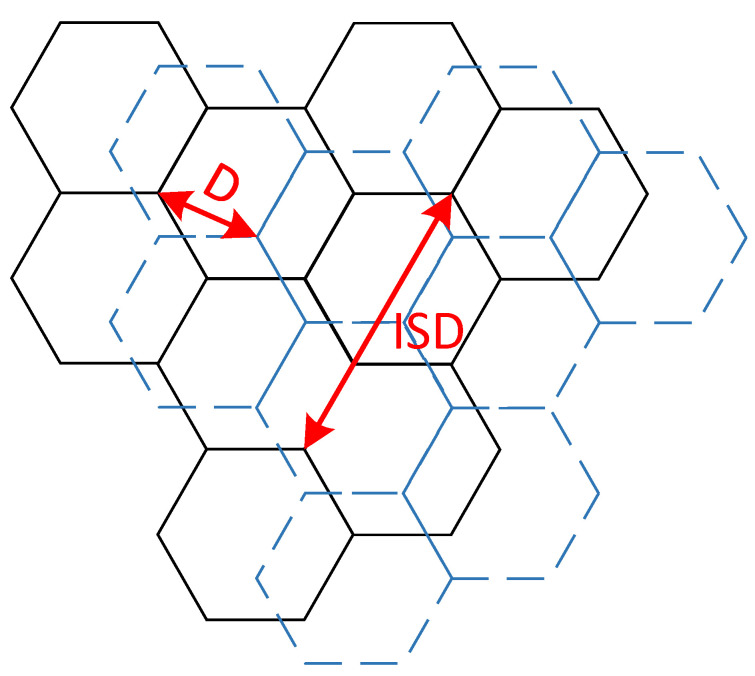
The offset between macro cellular networks.

**Figure 11 sensors-22-03876-f011:**
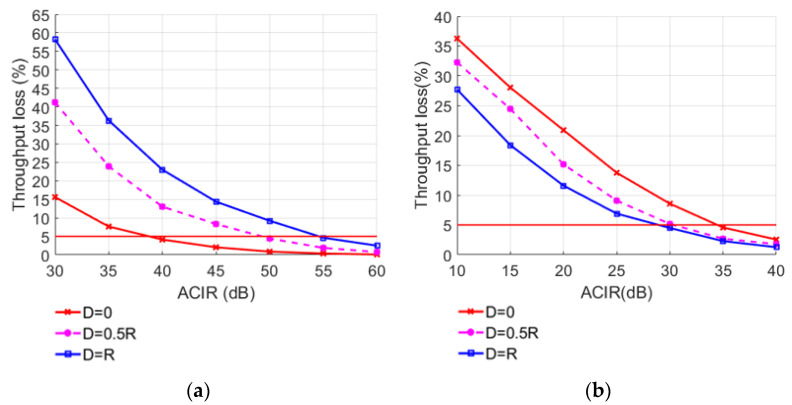
LTE-M throughput loss under different location offset between base stations and ACIR: (**a**) UE-BS, (**b**) UE-TE.

**Figure 12 sensors-22-03876-f012:**
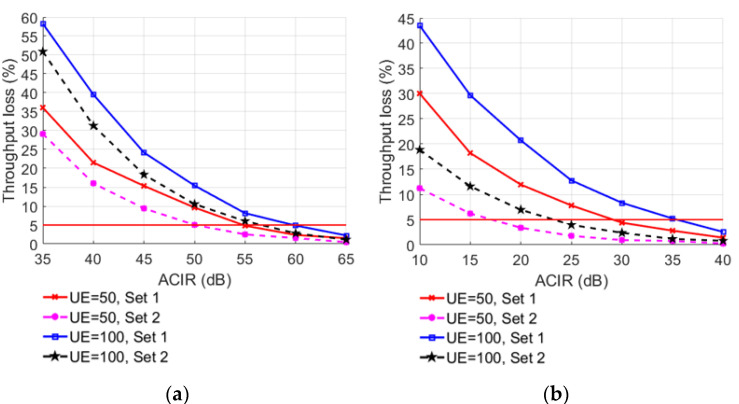
LTE-M throughput loss under different number of us ers, ACIR, and power control parameter sets: (**a**) UE-BS, (**b**) UE-TE.

**Table 1 sensors-22-03876-t001:** LTE-M band with upper and lower adjacent frequency range.

Interference Band	Frequency	Type of Interfering Link	Mode	Interference Mobility
Lower adjacent frequency	1765–1785 MHz	Uplink signal	LTE FDD	Movable
Upper adjacent frequency	1805–1820 MHz	Downlink signal	LTE FDD	Immovable
	1820–1825 MHz	Downlink signal	GSM FDD	Immovable

**Table 2 sensors-22-03876-t002:** Interference calculation parameters of LTE-M with directional antenna radiations pattern.

Interference Link	UE-BS				UE-TE			
Received BW (MHz)	5		10		5		10	
Guard band (MHz)	0	5	0	5	0	5	0	5
$P_{Imax}$ (dBm)	23		23		23		23	
$G_{TR}$ (dBi)	15		15		0		0	
$P_{noise}$ (dBm)	−102		−99		−98		−95	
$I_{threshold}$ (dBm)	−109		−106		−98		−95	
ACS (dB)	53.27		50.26		49.27		46.26	
ACLR (dB)	21.37	35.02	21.19	35.02	21.37	35.02	21.19	35.02
ACIR (dB)	21.37	34.96	21.19	34.89	21.36	34.86	21.18	34.71
MCL (dB)	125.63	112.04	122.81	109.11	99.64	86.14	96.82	83.29

**Table 3 sensors-22-03876-t003:** Spectrum emission mask.

Spectrum Emission Limit (dBm)/Channel Bandwidth							
Δf_OOB_ (MHz)	1.4 MHz	3.0 MHz	5 MHz	10 MHz	15 MHz	20 MHz	Measurement Bandwidth
±0–1	−10	−13	−15	−18	−20	−21	30 kHz
±1–2.5	−10	−10	−10	−10	−10	−10	1 MHz
±2.5–5	−25	−10	−10	−10	−10	−10	1 MHz
±5–6		−25	−13	−13	−13	−13	1 MHz
±6–10			−25	−13	−13	−13	1 MHz
±10–15				−25	−13	−13	1 MHz
±15–20					−25	−13	1 MHz
±20–25						−25	1 MHz

**Table 4 sensors-22-03876-t004:** LTE FDD UE interfering with LTE-M BS.

Location of UE	Platform				Carriage			
Received BW (MHz)	5		10		5		10	
Guard band (MHz)	0	5	0	5	0	5	0	5
Minimum safe distance (m)	25.87	5.41	18.70	3.86	10.30	2.15	7.44	1.54

**Table 5 sensors-22-03876-t005:** LTE FDD UE interfering with LTE-M TE.

Location of UE	Platform				Carriage			
Received BW (MHz)	5		10		5		10	
Guard band (MHz)	0	5	0	5	0	5	0	5
Minimum safe distance (m)	1.30	0.28	0.94	0.20	0.52	0.11	0.37	0.08

**Table 6 sensors-22-03876-t006:** Interference calculation parameters of LTE-M with leaky cable radiations pattern.

Interference Link	UE-BS			
Received BW (MHz)	5		10	
Guard band (MHz)	0	5	0	5
$P_{Imax}$ (dBm)	23			
Attenuation factor a (dB/100 m)	4			
Slot period p (m)	0.25			
$\mathrm{Coupling}\;\mathrm{loss}\;\mathrm{at}\;2\;m\;L_{2m}$ (dB)	62			
Feeder loss (dB)	0.9			
Splitter loss (dB)	1.4			
Train penetration loss (dB)	8			
Body penetration loss (dB)	2			
$I_{threshold}$ (dBm)	−109		−106	
ACIR (dB)	21.37	34.96	21.19	34.89
MCL (dB)	110.63	97.04	107.81	94.11
Minimum safe distance (m)	0.85	0.18	0.61	0.13

**Table 7 sensors-22-03876-t007:** ACLR model for LTE interfere and victim.

LTE BW	Number of RBs per UE	$\mathrm{Interference}\;\mathrm{BW}\;B_{A}$	$\mathrm{ACLR}\;\mathrm{with}\;\mathrm{Interval}\;\mathrm{Less}\;\mathrm{than}\;B_{A}$	$\mathrm{ACLR}\;\mathrm{with}\;\mathrm{Interval}\;\mathrm{Greater}\;\mathrm{than}\;\;B_{A}$
5 MHz	5	5 × 180 KHz	30 + X	43 + X
10 MHz	10	10 × 180 KHz	30 + X	43 + X
20 MHz	20	20 × 180 KHz	30 + X	43 + X

**Table 8 sensors-22-03876-t008:** Power control parameter sets.

Parameter Set	Gamma (*γ*)	*PL_x-ile_* (dB)			
		20 MHz BW	15 MHz BW	10 MHz BW	5 MHz BW
Set 1	1	109	110	112	115
Set 2	0.8	N/A	N/A	129	133

**Table 9 sensors-22-03876-t009:** LTE link performance baseline.

Parameter	Downlink	Uplink
α	0.6	0.4
${\mathrm{SINR}}_{min}/\mathrm{dB}$	−10	−10
${\mathrm{TP}}_{max}/\mathrm{bps}\cdot {\mathrm{Hz}}^{-1}$	4.4	2.0
${\mathrm{SINR}}_{max}/\mathrm{dB}$	22.05	14.91

**Table 10 sensors-22-03876-t010:** Simulation parameters of the proposed system.

Parameters	LTE FDD		LTE-M	
	Uplink	Downlink	Uplink	Downlink
Cell Structure	Macrocell structure, 750 m distance between base stations		Linear topology	
Carrier BW	10 MHz		10 MHz	
RB Size	180 KHz		180 KHz	
User distribution	Uniformly distribute at random based on area		Nearby random distribution along the track	
User/Train Number	50,100		1	
Thermal noise density	−174 dBm/Hz		−174 dBm/Hz	
Noise coefficient	5 dB	9 dB	5 dB	9 dB
Antenna Height	6 m	1.5 m	5 m	2.5 m
Receiving Antenna Gain	15 dBi	0 dBi	15 dBi	0 dBi
Transmitting Antenna Gain	0 dBi	15 dBi	0 dBi	15 dBi
Maximum/Minimum Transmitting Power	23 dBm/−30 dBm	43 dBm	33 dBm/−30 dBm	46 dBm

**Table 11 sensors-22-03876-t011:** The value of ACIR (dB) at 5% throughput loss.

Interference Link	Offset D			Number of UE/Set			
	0	0.5R	R	50/Set 1	50/Set 2	100/Set 1	100/Set 2
UE-BS	38.76	49.25	55.08	50.07	54.82	56.61	59.79
UE-TE	28.94	30.35	34.36	17.15	23.26	29.12	34.86

## Data Availability

Not applicable.

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
