# Peer review of "Analysis of LTE-M Adjacent Channel Interference in Rail Transit"

_sensors, 2022, doi:10.3390/s22103876_

Round 1

Reviewer 1 Report

The paper contains an analysis of adjacent channel interference in rail transit on LTE-M system. For that, authors provided a Monte Carlo simulation to analyse several conceivable scenarios of the LTE-M system and the adjacent frequency cellular system.

The importance and timeliness of the topic addressed in the paper within its area of research is good.

The structure of the paper is clear, but it makes the results of this work not clear. In particular, the section 3 and 4 define two different analysis with two different results. I suggest to create one section with methodology used and one section with results obtained. In this way, it is possible to compare the results obtained in underground with ground scenario.

The analysis of litterature review is not complete, there are only 16 references of others papers. I suggest to increase the number of references, due to the manuscript contains a long description of challenges of this analysis.

The originality and novel results are not highlight in this work. I suggest to increase the importance of manuscript motivation in the text.  

Reviewer 2 Report

0) The paper considers thee important point of operation of LTE technology in a railway context for signaling purposes. What is missing is a better characterisation of the railway environment and of performance requirements for signaling applications. The risk is considering the LTE-M system (and similar protocols) with criteria bounded and suitable for commercial applications (e.g. voice, internet, entertaining).

Please, besides this general comment, consider the following more detailed comments 1) to 11).

 1) Line 29. Ministry of which country? is it a regulation that is internationally relevant?
2) Line 32. A starting point is compliance to ETSI standards. There are requirements for channel separation, channel-to-channel interference and spurious emissions control. Thy must be included in your Introduction and the rest of the paper where necessary, to justify the work and provide the grounding for what proposed.
3) Line 50. At lines 32-50 you have listed literature that has considered mutual interference between telecommunications services. What are the outcomes? is it critical? under which circumstances?
4) At line 51 you introduce your work and justify it on the ground that at 1.8 GHz it has not been studied yet. Please, highlight what are the expected difficulties and differences that impede translating results from other studies into the considered scenario.
5) Section 3. You state that the source of interference is LTE FDD equipment, but you should not forget that the metro and railway environment is unfavorable for the presence of sources of electromagnetic noise, that superpose to interference from other telecom systems. Please, justify how you took these phenomena into account. There are some literature references quantifying disturbance from electric arc dynamics and in general problems of measurement and modeling of such emissions with respect to telecommunication systems.
6) Line 174. Are the considered models suitable for propagation in tunnels? you are speaking of building roof height, whereas for metros antennas are low on ground, aiming at tunnels and along track and viaducts; high towers and buildings are used for railways telecommunication systems and longer distances (e.g. GSM-R), or maybe to cover depots and shunting yards.
7) Measuring units should not be in italic. It seems the paper was prepared with Latex : in this case you can use package siunitx and \SI{}{} or simply \mathrm{} in formulas.
8) Table 5 - Train penetration loss. The LTE-M signal is for antennas on roof top, but not inside the train. Please, clarify.
9) Line 355. Isn't 5% reduction of throughput an extreme case? Usually FER and BER (that could cause packet loss and thus retransmission and reduction of throughput) are at 0.1% or better. Regarding availability and safety for metro and railway applications you should refer to standards such as those for CBTC or GSM-R. At least as a starting point that has had a long trial period.
10) Table 9. Again on antenna height. Who places antennas at that height for metro applications? Could you please substantiate this choice with e.g. system topology, covered sectors, number of trains getting in and out, and number of occupied channels?
11) References. Not in MDPI style. They should also be increased in number to include discussion of propagation in tunnels and typical noise affecting telecommunications in railways and metros, such as for GSM-R and train wayside radio links, as well as characterization of the electromagnetic environment, electromagnetic emissions and electromagnetic compatibility related to rolling stock and railways.

Reviewer 3 Report

The authors work on a very interesting and promising research area. Please consider the following remarks:

1)  Please provide references for Section 2.

2) At lines 202, 264, 271 insert the word "equations" in text.

3) At lines 118, 124, 147, 150 change the specification number with just the reference number.

4) Please provide the main research findings at Abstract section.

5) Please explain each abbreviation the first time it appears in text (e.g. UE and TE).

6) Which software tool has been used to obtain the simulation results? Please explain. 

7) Reference list is adaquate and up to date.

Round 2

Reviewer 1 Report

New manuscript version is more clear than previous version, authors have followed the reviewers suggestions.

Reviewer 2 Report

Dear Authors,

I once more congratulate for this work: I think it's very interesting and thank you for the extensive amendments.

(as for point 7, I thought you used Latex because the pdf is very accurate for line numbering and few other elements)